# Residents' Engagement Behavior in Destination Branding

**Yuzong Zhao** ⓘ, **Xiaotian Cui and Yongrui Guo** *ⓘ

School of Tourism and Geography Science, Qingdao University, 308 Ningxia Road, Qingdao 266071, China;
zhaoyuzong@126.com (Y.Z.); 15165840312@163.com (X.C.)
* Correspondence: guoyongrui58@163.com; Tel.: +86-178-0504-7532

**Abstract:** Residents play an important role in the destination branding process. Extant studies have not yet integrated analyses of residents' engagement behavior and the factors that affect it. In this study, we investigated the influence of place identity, place brand identity, place brand commitment, and perceived benefits of tourism on residents' engagement behavior (brand ambassadorship behavior, brand citizenship behavior) in destination branding. Derived from a sample of 380 residents of Zhouzhuang, one of China's most important tourism destinations, the data for this research were analyzed using partial least squares structural equation modeling (PLS-SEM) using SmartPLS 3.3.2. The findings reveal that residents' emotions towards destinations and brands and the perceived benefits of tourism significantly positively influence brand ambassadorship behavior and brand citizenship behavior. Residents' engagement behavior in destination branding is influenced by mechanisms of social identity and social exchange. This study reveals the antecedents that affect residents' engagement behavior in destination branding. The results also provide comprehensive insight into residents' engagement behavior in destination branding based on social identity and social exchange theories.

**Keywords:** destination branding; place branding; brand ambassadorship behavior; brand citizenship behavior; brand identity; place identity

## 1. Introduction

Branding has existed for centuries, and the importance of branding seems unlikely to change and has become increasingly important [1]. Brands can reduce perceived risks, set expectations, and simplify consumer decision making. Destination branding became a topic of interest in the late 1990s and has gained increasing attention in recent years [2–4]. From the perspective of destinations, branding creates brand equity [5], strengthens differences among competitors, and enhances tourism competitiveness [6,7]. From the perspective of tourists, branding affects tourist behavior and emotions towards destinations, such as word of mouth, loyalty, and satisfaction [8,9]. From the perspective of residents, branding influences residents' place identification, place attachment, and support for tourism [10–12]. Thus, destination branding is a powerful instrument to achieve a competitive advantage that allows destinations to increase their attractiveness to tourists and enhance the place satisfaction of residents.

Although our understanding of destination branding has grown tremendously, there has been less attention given to understanding stakeholder engagement in the destination branding process. The true nature of place/destination branding is the interaction and dialogue between stakeholders [13]. Destination brands are embedded in the local society and culture and co-created and unified by social actors [14]. As key stakeholders in a destination system, residents' roles must be identified and considered [15,16]. Residents are informal, authentic, and reliable sources of destination marketing, and they are also active proponents of destination branding [17,18]. Residents must transform from passive targets to active co-creators in the destination branding process.

In the field of place branding research, researchers have found that emotional connections to a place affect residents' place brand engagement behavior. Residents' emotional commitment, place attachment, and satisfaction with the place are the most important goals of place marketing, and represent the key to the success of place branding [19,20]. However, there are differences between place branding and destination branding. Destination branding has a more market-oriented and top-down formulation [21] that is not closely related to residents because residents are not involved in the process of destination branding [15]. For a long time, the pursuit of economic interests and the resulting attention to the interests of tourists have caused destination branding to mainly focus on the external orientation. In addition, residents are the most affected stakeholders in destination branding. Residents can play diversified roles in destination branding, such as spreading positive word of mouth. Therefore, residents' brand attitudes and the perceived benefits of tourism have important influence in destination branding. Nevertheless, extant studies have not yet integrated analyses of the factors affecting residents' engagement behavior in destination branding.

To fill this research gap, in this study, we systematically examined the relationship between residents' place attitudes, brand attitudes, perceived benefits of tourism, and destination brand engagement behavior. The research objectives of this study are twofold: (1) to examine whether residents' emotions towards destinations and brands and perceived benefits of tourism influence brand engagement behavior, and (2) to investigate whether this influence is different in the two dimensions of brand engagement behavior, brand ambassadorship behavior and brand citizenship behavior. This research reports the results of an empirical analysis of a leading destination in China to test the proposed hypotheses.

This study contributes to the body of knowledge about residents' engagement behavior in destination branding in two ways. First, this study reveals the antecedents that affect residents' engagement behavior in destination branding. Second, this study contributes comprehensive insight to understand residents' engagement behavior in destination branding based on social identity and social exchange theories.

## 2. Literature Review and Hypotheses Development

### 2.1. Residents' Role in Destination Branding

The increasingly competitive global tourism industry has pushed destinations to use branding as a way of achieving competitive advantage [22]. Like any brand, destinations must be able to define where their unique attractiveness lies. The strategic goal of destination branding is to strengthen place identity and differentiation from others by selecting a consistent local element mix [6].

A brand is a relationship with consumers and other stakeholders [23]. Interactions and dialogues between stakeholders are considered the true nature of destination branding [13]. However, the destination branding process pays more attention to external stakeholders (tourists) and ignores internal stakeholders (residents). In other words, destination branding focuses more on "how others see us" rather than "how we see ourselves". There are some possible explanations for destination branding from a tourist point of view. From a theoretical perspective, destination branding research follows customer-based product brand equity theory. From a practical perspective, destination branding is a market-oriented strategy. From the perspective of the brand building process, a top-down brand strategy that reflects the interests of powerful stakeholders is prevalent [24].

The residents of a destination are potentially the largest and most powerful stakeholders of destination brands [16] and play an important role in shaping a competitive brand for their destination [25]. Kavaratzis (2012) stated three reasons for residents' participation and involvement in place branding [26]. The first is that place branding needs public support as a public management activity. The second reason is the recent turn towards participatory branding, which emphasizes the importance of internal audiences in general. The third reason is that the online world reinforces the importance and necessity of place brand communication channels. Uchinaka et al. (2019) also note that residents are primary

sources of place marketing and active proponents of place branding in the digital age [18]. Casais and Monteiro (2019) conclude that residents are the main agents who generate place authenticity and manifest the identity of a place by living and communicating with external parties [27].

Braun et al. (2013) proposed that residents are not only citizens and brand ambassadors of a place brand but also an integral part of the place brand [28]. The characteristics and behavior of residents are the constituent elements of place brands and the source of their message. Research by Chen et al. (2018) has shown that residents of destinations can act as destination brand ambassadors [29]. Uchinaka et al. (2019) identified four roles of residents as online place ambassadors: Contributor, Photographer, Hobbyist, and Retweeter [18]. Hudson et al. (2017) claimed that residents should be actively involved in place branding and take on the role of marketers and informed ambassadors of their place brand [30].

### 2.2. Residents' Engagement Behavior in Destination Branding

Based on a literature review, residents' engagement behavior in destination branding in this research includes two categories: brand ambassadorship behavior and brand citizenship behavior. Brand ambassador behavior involves residents' recommendation and promotion behavior to outsiders. Brand citizenship behavior is internal brand management behavior.

Taecharungroj (2016) defined city ambassadorship behavior as the behavior of residents who promote the city through positive word-of-mouth recommendations and communications [31]. Braun et al. (2013) also believe that the word-of-mouth behavior of residents who recommend their area to others is an important form of brand ambassadorship behavior [28]. Residents' word-of-mouth behavior is perceived as authentic and trustworthy, which has an important impact on tourists' destination choices. Wassler et al. (2019) suggested that brand ambassadorship behavior is considered to be potentially promotional and/or development-related and may occur in a planned way (e.g., by ambassadorial networks) or spontaneously (e.g., by unorganized communication) [16]. Previous research has identified some factors that have a positive impact on the brand ambassador behavior of residents. Taecharungroj (2016) found that resident satisfaction, identity, and commitment to the city positively influence city ambassadorship behaviors [31]. Chen et al. (2018) found that different dimensions of place attachment have an impact on different types of word of mouth [29]. Vollero et al. (2018) found that environmental attitudes, community commitment, and perceptions of the effectiveness of existing place marketing communications affect residents' engagement in the promotion of destinations [20]. When residents are satisfied and identify with a place and when they commit to the place, they can better enhance the destination experience and become living ambassadors of that place.

Brand citizenship behavior refers to spontaneous and voluntary brand-building behavior [32]. Taecharungroj (2016) defined city citizenship behaviors as "the behaviors of residents that contribute to the city by helping other people and participating in events that can improve the city" [31]. King et al. (2013) note that brand citizenship behavior is behavior that is beyond that formally prescribed but essential for brand promise [33]. Brand citizenship behavior is also manifested as employees' willingness to make extra efforts that exceed their basic duties and show brand-consistent behavior [34]. Brand citizenship behavior emphasizes that residents' brand behavior is a spontaneous, out-of-role behavior that is not evaluated by a formal reward and punishment system. Burmann and Zeplin (2005) outlined seven types of brand citizenship behavior: brand consideration, helping behavior, sportsmanship, brand enthusiasm, brand advancement, self-development, and brand endorsement [32]. In another study, Burmann et al. (2009) developed a brand citizenship behavior scale and conducted an empirical analysis, and the seven originally defined dimensions of brand citizenship behavior were reduced to three dimensions: brand enthusiasm, willingness to help, and propensity for further development [35]. Residents are the most important "cell" that defines and completes the brand of a destination [36]. Residents' engagement in destination branding is a form of citizenship with duties and

responsibilities to promote place development [37]. Residents' support and advocacy play a vital role in the destination branding process.

*2.3. Place Identity*

According to social identity theory, social identity is the feeling that the individual is part of the collective group. According to Ashforth and Mael (1989) [38], identity has three important effects on organizations. First, people tend to choose activities that are consistent with their identities, and they support the organizations that embody those identities. Second, identity affects important organizational outcomes. Third, the more employees identify with the organization, the more distinctive and unique the values and practices of the organization become. Place identity is a specific subtype of social identity and involves the ways in which the physical and symbolic attributes of place shape an individual's sense of self or identity. Identity and differentiation are considered to be the two important functions of destination branding [7]. Destinations are embedded in place, and place identity is rooted in and creates the uniqueness of place characteristics. Destination brands should portray an attractive and distinctive image that highlights the core culture and identity of a place [39].

As an emotional connection with a place, place identity has been explored to predict place-related behaviors. Zenker and Rütter (2014) found that citizens with higher levels of satisfaction were more likely to talk positively about their city [40]. Zhang and Xu (2019) found that destination psychological ownership positively influences place attachment, which further influences place citizenship behavior [41]. Guo et al. (2018a) confirmed that place identity improves residents' perceived resilience in tourism destinations [42]. Stylidis (2018) reported that place attachment and support for tourism increase the positive word of mouth of residents [10]. Residents' place identity is regarded as both an aim and facilitator of destination branding [11]. Based on the above discussion, we examine the following hypotheses:

**Hypotheses 1 (H1).** *Place identity has a positive effect on residents' brand ambassadorship behavior.*

**Hypotheses 2 (H2).** *Place identity has a positive effect on residents' brand citizenship behavior.*

*2.4. Place Brand Identity*

Brand identity is central to a brand's strategic vision. In branding research, brand identity refers to the perception of sameness and oneness between the brand and the consumer [43,44]. From a tourist perspective, brand identity is a psychological state in which tourists evaluate the belongingness between themselves and the destination brand [45]. Previous research on brand identity focused more on consumers and adopted a demand perspective. Understanding brand identity from a supply-side perspective can provide new insights for practitioners. Residents are integral to the destination brand, and brand identity should be derived from the identities held by residents [46].

Scholars recognize that brand identity has a positive effect on brand building behaviors. Kuenzel and Halliday (2008) confirmed that consumers' development of relationships via brand identity results in intentions to repurchase the brand and encourages word of mouth about the brand [47]. Nam et al. (2011) suggested that brand identity has positive effects on consumer satisfaction in the hotel and restaurant industries [48]. Kemp et al. (2012) found that in the branding and positioning of a place, residents' attitudes towards branding activities are very important [49]. In a similar vein, residents who identify with their own destination brands are more likely to show hospitable attitudes and behaviors towards tourists [50]. Based on the discussion above, we examine the following hypotheses:

**Hypotheses 3 (H3).** *Place brand identity has a positive effect on residents' brand ambassadorship behavior.*

**Hypotheses 4 (H4).** *Place brand identity has a positive effect on residents' brand citizenship behavior.*

### 2.5. Place Brand Commitment

Brand commitment reflects an individual's psychological and emotional attachment to a brand. Brand commitment affects the extra effort that individuals are willing to make to achieve brand goals [32]. Commitment and identity are two closely related but distinct concepts. Commitment refers to the relationship between separate psychological entities, whereas identity reflects psychological oneness [19]. In place branding research, Ahn et al. (2016) defined place brand commitment as an individual's psychological attachment to a place brand [15]. Coelho et al. (2020) note that the commitment to a brand is the psychological connection with the place brand [51]. Residents' place brand commitment plays an important role in improving internal brand management procedures [35].

Residents with affective place brand commitment will have positive spontaneous behavior [52]. Brand commitment is a necessary condition for the successful strengthening of a brand [35]. Kemp et al. (2012) clarified and tested the relationships between brand commitment and self-brand connections [49]. Their study results indicate that brand commitment, attitude, trust, and uniqueness are crucial to building self-brand connections among residents. Xiong et al. (2013) concluded that brand commitment is a strong predictor of hospitality employees' pro-brand behavior, that is, brand endorsement, brand allegiance, and brand consistent behavior [53]. Previous marketing literature has found that place brand commitment positively affects the word-of-mouth behavior of residents [54,55]. Based on the above discussion, we examine the following hypotheses:

**Hypotheses 5 (H5).** *Place brand commitment has a positive effect on residents' brand ambassadorship behavior.*

**Hypotheses 6 (H6).** *Place brand commitment has a positive effect on residents' brand citizenship behavior.*

### 2.6. Perceived Benefits of Tourism

Residents' perceived benefits of tourism have been widely discussed in tourism literature. Tourism increases household income and improves the standard of living, increases employment opportunities for residents, protects the environment, and produces new cultural and entertainment activities. An accumulating body of evidence demonstrates that the positive benefits associated with tourism activities have a positive impact on tourism development [56]. The perceived benefits of tourism positively affect residents' support for destination development [57], trust in government actors [58], overall life satisfaction [59], community participation [60], community resilience [61], and word-of-mouth behaviors [29]. According to the theory of social exchange, when residents' perceived benefits of tourism are greater than the costs paid, they will show a positive attitude. In turn, positive attitudes and satisfaction can transform residents into a destination's most valuable ambassadors [16,62]. In the management discipline, previous literature has found that high levels of benefits and satisfaction usually mean very positive destination image and brand perceptions, positively affecting citizenship behaviors [15].

Based on the above discussion, we examine the following hypotheses:

**Hypotheses 7 (H7).** *The perceived benefits of tourism have a positive effect on residents' brand ambassadorship behavior.*

**Hypotheses 8 (H8).** *The perceived benefits of tourism have a positive effect on residents' brand citizenship behavior.*

The conceptual model is provided in Figure 1.

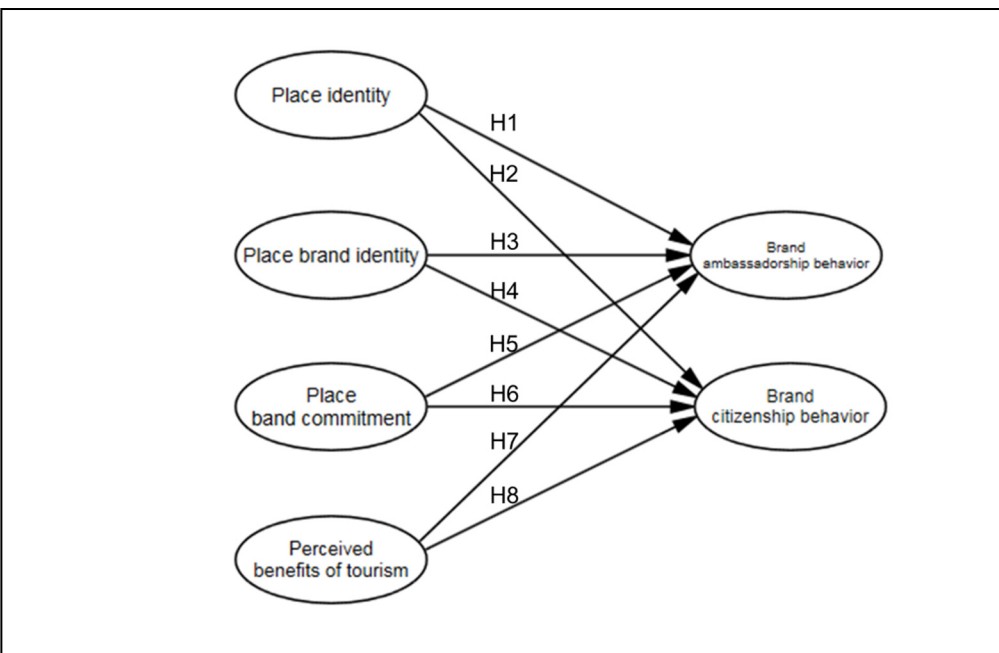

**Figure 1.** Conceptual model.

## 3. Methodology

### 3.1. Measurement of Variables

All constructs were assessed by a 5-point Likert scale (1 = strongly disagree, 5 = strongly agree). Place identity was measured by four items adapted from Williams and Vaske (2003) [63]. The place brand identity measure included five items from Zenker et al. (2017) [11] and Choo et al. (2011) [50]. A four-item scale to measure place brand commitment was applied and modified from the study by Ahn et al. (2016) [15]. The scale related to brand ambassadorship behavior consisted of four items adapted from Taecharungroj (2016) [31] and Chen and Dwyer (2018) [64]. Brand citizenship behavior was assessed by four items also adapted from Taecharungroj (2016) [31] and Chen and Dwyer (2018) [64]. The perceived benefits of tourism were measured via a 12-item scale derived from Stylidis et al. (2014) [65] and Gursoy and Rutherford (2004) [66].

### 3.2. Data Collection and Sample Profile

Zhouzhuang, located in Suzhou, China, was chosen as the study site. Zhouzhuang is famous for its beautiful water scenery, well-preserved ancient residential houses, and profound cultural background. The town covers an area of 38.96 square kilometers, of which the water area is 18.16 square kilometers. In 1983, the famous painter Chen Yifei's painting "Double Bridge" made Zhouzhuang known to the world. In 1989, Zhouzhuang began to develop tourism. It enjoys a reputation worldwide as "China's first water town". It is among the first Chinese historical and cultural towns and among the first national AAAAA (5A) tourist attractions. Zhouzhuang is committed to building a strong tourism destination brand. In recent years, "Better lifestyle, better life in Zhouzhuang" has become Zhouzhuang's tourism brand image. Zhouzhuang is currently one of the most well-known tourist destination brands in China.

Six undergraduates and four graduate students majoring in tourism management collected data. To ensure the quality of the survey, all investigators were scientifically trained. All of the investigators needed to know the background information of the destination, the goals of the survey, and the content of the questionnaire. Participation in this study was voluntary. Since most of the residents of Zhouzhuang are engaged in tourism, to ensure that the respondents had sufficient time to fill out the questionnaire, the data collectors chose to work in the morning when there were fewer tourists. The data

were collected from 11 to 18 July 2019. As a result, 400 questionnaires were returned, 380 of which were valid.

As shown in Table 1, of the 380 respondents, 45.8% were male and 54.2% were female. The represented age categories included 18 to 35 (48.7%), 36 to 49 (38.2%), and 50 years of age or older (13.2%). The level of education ranged from less than high school (38.1%) to high school/vocational school (29.2%) to an undergraduate degree and above (32.6%). There are a large number of lifestyle tourism migrants in Zhouzhuang, and native residents account for only 59.7%. The most common type of tourism business in Zhouzhuang is restaurants (25.2%), followed by souvenir shops (22.9%) and hotels (20.0%). The average annual personal income from tourism is RMB 81,000.

**Table 1.** Sample characteristics.

| Attribute | n | 100% |
|---|---|---|
| **Gender** | | |
| Male | 174 | 45.8 |
| Female | 206 | 54.2 |
| **Age** | | |
| 18–35 | 185 | 48.7 |
| 36–49 | 145 | 38.2 |
| 50 or older | 50 | 13.2 |
| **Education** | | |
| Less than high school | 145 | 38.1 |
| High school/vocational school | 111 | 29.2 |
| Undergraduate degree and above | 124 | 32.6 |
| **Residents' characteristics** | | |
| Native residents | 227 | 59.7 |
| Lifestyle tourism migrants | 153 | 40.3 |
| **Business type** | | |
| Restaurants | 96 | 25.2 |
| Souvenir shops | 87 | 22.9 |
| Hotels | 76 | 20.0 |
| Other tourism business | 121 | 31.8 |

*3.3. Data Analysis*

In this study, we used partial least squares structural equation modeling (PLS-SEM) to analyze the conceptual model. PLS-SEM relaxes normal distributional assumptions and is used specifically in testing path model hypotheses in an exploratory manner. We tested for multivariate normality by the Kolmogorov–Smirnov test. The results indicated a non-normal data distribution. Thus, PLS-SEM was an appropriate method for this study.

## 4. Results

*4.1. Non-Response Bias*

According to the recommendations of Armstrong and Overton (1977) [67], we conducted a non-response bias test of the questionnaire using SPSS 22 statistical software. First, we divided the questionnaire into two parts according to the time sequence of return: early responders (the first 25% of the questionnaires) and late responders (the last 25% of the returned questionnaires). Second, we compared the two groups by chi-square tests. The results showed that there were no significant differences in the control variables of gender and age between the two groups at the 5% confidence interval. Therefore, this study excluded the possibility of non-response bias.

*4.2. Common Method Bias*

In this study, potential common method bias was assessed by Harman's single-factor test. This method loads all items into an exploratory factor analysis, examines the results of unrotated factor analysis, and determines the minimum number of factors necessary to explain the variance of the variable. If only one factor is extracted, or a factor has a very large explanatory power, it can be judged that there is a serious common method bias. The results show that no general factor accounted for more than 50% of the variance, the first factor accounted for 43.2%, and the combined six factors accounted for 68.9% of the total variance. Therefore, there is no common method bias in this study.

*4.3. Measurement Model*

To validate the measurement model, factor loadings, reliability, discriminant validity, and convergent validity were assessed. We deleted four items with factor loadings lower than 0.7 to improve the validity of the constructs. The measurement items for each construct and descriptive statistics of these measurements are presented in Table 1. Reliability assessment depends on Cronbach's $\alpha$ and the composite reliability (CR) estimate of each construct. In this research, all the Cronbach's $\alpha$ values fell between 0.770 and 0.918, and the CR values were between 0.867 and 0.936. Convergent validity was assessed using the average variance extracted (AVE) for each construct. The AVE value of all constructs exceeded 0.5. These results suggest that the measurement model is reliable and valid.

The heterotrait/monotrait (HTMT) ratio of correlation [68] and the Fornell–Larcker criterion [69] are used to evaluate discriminant validity. As shown in Table 2, the values of HTMT do not exceed the required threshold value of HTMT.90 by Gold et al. (2001) [70]. In the case of the Fornell–Larcker criterion, the results show that the square root of each construct's AVE is higher than the correlations with other constructs (see Table 3). The results indicate that discriminant validity was achieved (see Table 4).

**Table 2.** Descriptive statistics, reliability, and validity of the constructs.

| Constructs and Items | Mean | SD | Loading | AVE | CR | Cronbach's $\alpha$ |
|---|---|---|---|---|---|---|
| *Place Identity* | | | | | | |
| I identify strongly with Zhouzhuang | 4.20 | 0.859 | 0.855 | | | |
| I feel Zhouzhuang is a part of me | 3.92 | 1.057 | 0.857 | 0.729 | 0.890 | 0.814 |
| Zhouzhuang is very special to me | 4.16 | 0.929 | 0.849 | | | |
| *Place Brand Identity* | | | | | | |
| If someone criticizes Zhouzhuang, it feels like he is criticizing me | 3.54 | 1.096 | 0.746 | | | |
| If someone talks positively about Zhouzhuang, it feels like a compliment | 3.84 | 1.021 | 0.837 | 0.591 | 0.878 | 0.826 |
| I am very interested in what others think about Zhouzhuang | 4.02 | 0.896 | 0.700 | | | |
| Zhouzhuang's brand image overlaps with my self-image | 3.63 | 1.024 | 0.813 | | | |
| I identify with Zhouzhuang's tourism destination brand | 3.80 | 1.013 | 0.740 | | | |
| *Place Brand Commitment* | | | | | | |
| I am proud of the success of the Zhouzhuang tourism brand | 3.93 | 0.965 | 0.863 | | | |
| I am a loyal supporter of the Zhouzhuang tourism brand | 3.92 | 0.985 | 0.917 | 0.787 | 0.936 | 0.909 |
| I will make an effort to develop the Zhouzhuang tourism brand | 4.20 | 0.855 | 0.889 | | | |
| I truly care about the tourism brand of Zhouzhuang | 4.02 | 0.965 | 0.879 | | | |
| *Perceived Benefits of Tourism* | | | | | | |

**Table 2.** *Cont.*

| Constructs and Items | Mean | SD | Loading | AVE | CR | Cronbach's $\alpha$ |
|---|---|---|---|---|---|---|
| Standard of living | 3.94 | 0.955 | 0.754 | | | |
| Household income | 3.90 | 0.962 | 0.763 | | | |
| Employment opportunities | 3.99 | 0.956 | 0.804 | | | |
| Investment opportunities | 3.86 | 0.993 | 0.760 | | | |
| Cultural revival | 4.11 | 0.913 | 0.743 | 0.577 | 0.932 | 0.918 |
| Heritage conservation | 4.15 | 0.901 | 0.709 | | | |
| Infrastructure | 4.11 | 0.879 | 0.730 | | | |
| Cultural activities/entertainment | 4.15 | 0.875 | 0.740 | | | |
| Cultural exchange between tourists and residents | 4.20 | 0.880 | 0.759 | | | |
| Positive impact on cultural identity | 4.22 | 0.843 | 0.826 | | | |
| *Brand Ambassadorship Behavior* | | | | | | |
| I am willing to recommend Zhouzhuang to people who seek my advice | 4.27 | 0.796 | 0.830 | | | |
| I would encourage other people to come to Zhouzhuang | 4.36 | 0.783 | 0.828 | 0.675 | 0.893 | 0.839 |
| In tourist contact situations, I ensure that my personal appearance is in line with the appearance of Zhouzhuang residents in my mind | 4.12 | 0.827 | 0.844 | | | |
| I am willing to engage in promotional initiatives for Zhouzhuang | 3.85 | 0.947 | 0.783 | | | |
| *Brand Citizenship Behavior* | | | | | | |
| I am ready to attend events that are not required but help Zhouzhuang | 3.98 | 0.864 | 0.851 | | | |
| I would attend meetings and give opinions that can improve Zhouzhuang | 4.05 | 0.832 | 0.821 | 0.685 | 0.867 | 0.770 |
| I will take the initiative to clarify others' misunderstandings about Zhouzhuang | 4.17 | 0.778 | 0.810 | | | |

**Table 3.** HTMT results.

| | 1 | 2 | 3 | 4 | 5 | 6 |
|---|---|---|---|---|---|---|
| 1. Brand citizenship behavior | | | | | | |
| 2. Brand ambassadorship behavior | 0.895 | | | | | |
| 3. Place brand commitment | 0.742 | 0.801 | | | | |
| 4. Place brand identity | 0.703 | 0.767 | 0.820 | | | |
| 5. Place identity | 0.706 | 0.698 | 0.756 | 0.729 | | |
| 6. Perceived benefits of tourism | 0.651 | 0.657 | 0.650 | 0.511 | 0.569 | |

**Table 4.** Discriminant validity.

| | 1 | 2 | 3 | 4 | 5 | 6 |
|---|---|---|---|---|---|---|
| 1. Brand citizenship behavior | **0.828** | | | | | |
| 2. Brand ambassadorship behavior | 0.722 | **0.822** | | | | |
| 3. Place brand commitment | 0.621 | 0.700 | **0.887** | | | |
| 4. Place brand identity | 0.570 | 0.646 | 0.722 | **0.769** | | |
| 5. Place identity | 0.563 | 0.578 | 0.652 | 0.604 | **0.854** | |
| 6. Perceived benefits of tourism | 0.550 | 0.580 | 0.595 | 0.456 | 0.494 | **0.759** |

Note: Values on the bolded diagonal are the square root of the AVE.

### 4.4. Structural Model

Before the hypothesis test, in order to avoid the estimation bias caused by collinearity in the results, we used the VIF values to examine the collinearity problem of the structural model. The VIF values of the inner model in this study were all lower than 3, so there was no collinearity in our data.

The assessment of the structural model was performed through a bootstrapping technique (5000 subsamples) to test the research hypotheses. To evaluate the conceptual model's predictive capability, Stone–Geisser's $Q^2$ value was used to test the predictive relevance. The $Q^2$ values were above the threshold in this study. The model fit criterion implemented for PLS-SEM was the standardized root mean square residual (SRMR). The SRMR value in this study was 0.067, which is less than the recommended value of 0.08. The model's explanatory power was evaluated by the coefficient of determination (R2) [71]. The $R^2$ also reflects the in-sample predictive power. The $R^2$ values of brand ambassadorship behavior and brand citizenship behavior were 0.576 and 0.482, respectively. These results indicate the high explanatory power of this model.

Place identity did not have significantly positive effects on brand ambassadorship behavior ($\beta = 0.111$, $p > 0.05$) but had significantly positive effects on brand citizenship behavior ($\beta = 0.186$, $p < 0.01$). Place brand identity had significantly positive effects on brand ambassadorship behavior ($\beta = 0.250$, $p < 0.001$) and brand citizenship behavior ($\beta = 0.189$, $p < 0.01$). Place brand commitment had significantly positive effects on brand ambassadorship behavior ($\beta = 0.314$, $p < 0.001$) and brand citizenship behavior ($\beta = 0.220$, $p < 0.01$). The perceived benefits of tourism had significantly positive effects on brand ambassadorship behavior ($\beta = 0.224$, $p < 0.001$) and brand citizenship behavior ($\beta = 0.241$, $p < 0.001$). Thus, H1 was not supported, and H2–H8 were supported (see Table 5).

**Table 5.** Results of the structural model and hypotheses testing.

| Hypotheses/Path | Standardized Estimate | *t*-Value | *p*-Value | Results |
|---|---|---|---|---|
| H1. Place identity → brand ambassadorship behavior | 0.111 | 1.904 | 0.057 | Not supported |
| H2. Place identity → brand citizenship behavior | 0.186 | 2.911 | 0.003 | Supported |
| H3. Place brand identity → brand ambassadorship behavior | 0.250 | 4.019 | 0.000 | Supported |
| H4. Place brand identity → brand citizenship behavior | 0.189 | 2.620 | 0.009 | Supported |
| H5. Place brand commitment → brand ambassadorship behavior | 0.314 | 4.074 | 0.000 | Supported |
| H6. Place brand commitment → brand citizenship behavior | 0.220 | 3.004 | 0.003 | Supported |
| H7. Perceived benefits of tourism → brand ambassadorship behavior | 0.224 | 4.325 | 0.000 | Supported |
| H8. Perceived benefits of tourism → brand citizenship behavior | 0.241 | 4.418 | 0.000 | Supported |

## 5. Discussion and Conclusions

### 5.1. Theoretical Implications

Previous destination branding research has predominately focused on tourists rather than residents. The academic literature on residents and destination brands has not delved into place brand ambassadorship behavior and brand citizenship behavior, and lacks an antecedent model of these variables. Focusing on destination residents, in this study, we developed an integrated model that examined place identity, place brand identity, place brand commitment, and the perceived benefits of tourism as antecedents and place brand ambassadorship behavior and brand citizenship behavior as outcomes.

The results show that residents with a high level of place identity are more likely to engage in brand citizenship behavior. Psychological ties are an important motivation that affects people's behavior. Residents are beneficiaries and active co-creators of destination tourism development. Residents with higher perceptions of place identity are more inclined to engage in activities that can help the development of the destination, give opinions that can improve the tourist experience, and clarify others' misunderstandings about the destination. Residents' place brand citizenship behavior is a form of citizenship with responsibilities and duties for the benefit of the place [72]. Our findings provide additional support for recent suggestions that effective place branding should be able to express the culture of a place, shape the place identity, and communicate a unique image to others.

We also found that place identity does not have a significant positive effect on brand ambassadorship behavior. Compared with brand citizenship behavior, which focuses on internal management behavior, brand ambassador behavior tends to be an external

promotion behavior. As Govers and Go (2009) [73] note, place identity derives from the historical, cultural, political, religious, and local knowledge of a given place. This indigenous identity of residents significantly affects behavior within the group. This is consistent with Chen et al.'s (2018) finding that no significant effects of place identity on one-to-many and many-to-many word-of-mouth behaviors were found in Shanghai and Sydney samples [29].

Further supporting the hypothesized relationship derived from social identity theory, the study results suggest that greater levels of place brand identity can improve residents' engagement behavior in destination branding. According to social identity theory, members of different social groups define their social identities based on their group categories, and the place brand has become an important catalyst for residents to obtain social identity. Place brand identity makes residents become psychologically attached to and care about destination brands, which motivates them to expend more voluntary efforts to maintain and build stronger destination brands. Recently, Insch and Stuart (2015) revealed the importance of brand identity and argued that a lack of brand identity is the key factor that influences residents' place brand disengagement [74]. Residents' low identity with a place brand decreases their commitment to and acceptance of the brand. Residents who strongly identify with the destination brand will think and act on behalf of the brand. They are willing to recommend destinations to people who seek their advice and engage in place-promotion initiatives.

Baxter et al. (2013) suggest that place identity is pluralistic and fluid and that place brand identity is unitary and rigid [46]. They also note that place brand identity should be refined, designed, and embedded in the place identity set. If the place brand identity is not selected and rooted in the identities held by residents, it will lead to negative resident brand behaviors. Zenker and Beckmann (2013) found that Hamburg residents have low identity with the Hamburg brand and even participate in public protests about place marketing activities [75]. In the case of Lijiang, a world heritage site in China, Xu and Ye (2018) found that "The Capital of Yanyu" image, dominated by outsiders and market forces, is not an expected or desired image for local residents [76]. Zhouzhuang's destination brand, "Better lifestyle, better life in Zhouzhuang", is rooted in Zhouzhuang's lifestyle and comes from the place identity of residents. Residents' identity with the brand image of Zhouzhuang positively influences their role as place citizens and brand ambassadors.

The empirical results also demonstrate that place brand commitment has a positive effect on residents' brand ambassadorship behavior and brand citizenship behavior. This conclusion is in line with the work of Ahn et al. (2016) [15], who conclude that brand commitment is an important antecedent of brand citizenship behavior. Destination brands represent not only attractions, services, and goods, but also culture, ideologies, and people. Residents embody the culture of the destination and are an integral part of cultural experiences [77]. Residents who are committed to the brand are proud of the development of the destination. They are more willing to support and care about the development of destination brands. Therefore, destination residents need to clearly understand the meaning and knowledge [78] of the destination brand and commit to supporting brand development.

Previous research on residents' engagement behavior in destination branding has mainly focused on residents' emotions and attitudes towards brands. However, residents have both psychological and functional connections to destination brands. Residents' engagement in destination branding is influenced by mechanisms of both social identity and social exchange. The perceived benefits of tourism have a significant positive effect on brand ambassadorship behavior and brand citizenship behavior. The exchange of benefits is the underlying basis for human behavior [79]. Residents engage in destination branding with the expectation that doing so will be rewarding. The branding strategy enhances the competitiveness of the destination, attracts more tourists, and promotes the development of the destination. Tourism development produces economic, environmental, and social benefits to the residents of the destination; improves the quality of life of residents; and

enhances their well-being. The primary motivation for residents to support destination brand development is to improve the social and economic well-being of the community.

### 5.2. Practical Implications

This study has implications for management practice. First, residents should be involved in all stages of the destination strategic brand management process. Destination brand positioning should involve expressions of place identity as represented by a place's history, economy, culture, and residents. In the destination brand marketing stage, the destination marketing organizations should emphasize the role of residents as brand citizens and brand ambassadors. The behavior of residents' brand citizens has the characteristics of spontaneity and consciousness, and the target is mainly tourists. It includes the dimensions of willingness to help, brand enthusiasm, and propensity for further development [35]. For example, residents take the initiative to introduce the destination brands and slogans to tourists, clarify their misunderstanding of brands and deliver positive information to tourists, and maintain brand image and reputation. Brand ambassadors have rich brand knowledge, high brand commitment, and positive brand behavior. The behavior of residents' brand ambassadors includes the brand recommendations and brand building, etc. For example, "Moon Grandma" in China can communicate with tourists in 11 languages, and spread Guilin destination brands through the WeChat platform, etc. Destination brand evaluation should take residents' satisfaction as an important evaluation indicator. Second, the relationship between residents and destination brands should be strengthened. On the one hand, destination marketing organizations need to actively guide residents to participate in the brand construction process. On the other hand, they should help residents become familiar with brand objectives, establish destination brand identity, understand destination brand meaning, elicit positive brand responses, and, ultimately, establish a loyal, active, and intense relationship between residents and the destination brand. More importantly, destination marketing organizations need to present the brand construction process in a way that is easy for residents to understand and act on. Third, the research results show that residents' engagement behavior in destination branding is directly affected by the perceived benefits of tourism. Government decision makers and tourism developers need to consider the needs of community residents, create more opportunities for residents to engage in tourism development, improve residents' quality of life and economic income, reduce environmental and social costs, increase overall satisfaction, obtain community support, and promote sustainable tourism development.

### 5.3. Limitations and Future Research

Although the findings of the present study contribute to the literature on destination branding, several limitations of the current study should be acknowledged. First, we examined our hypotheses by taking a case grounded in China's destination brand management mechanism, which emphasizes top-down place branding schemes. Future studies can consider the model provided in other cultural settings. Second, the sample in this study did not distinguish between local residents and temporary residents. Temporary residents, such as lifestyle tourism migrants, differ from local residents in terms of place identity and place brand identity. This indicates a direction for future research.

**Author Contributions:** Conceptualization, Y.Z. and Y.G.; Data curation, X.C.; Formal analysis, Y.Z.; Funding acquisition, Y.Z. and Y.G.; Investigation, Y.Z., X.C., and Y.G.; Methodology, X.C. and Y.G.; Project administration, Y.G.; Writing—original draft, Y.Z. All authors have read and agreed to the published version of the manuscript.

**Funding:** This research was funded by the National Natural Science Foundation of China (grant no. 41871126, 41801135, 41801144) and the National Social Science Foundation of China (grant no. 19BJY215).

**Institutional Review Board Statement:** Approval for the study was not required in accordance with local/national legislation.

**Informed Consent Statement:** Informed consent was obtained from all subjects involved in the study.

**Data Availability Statement:** The data presented in this study are available on request from the corresponding author. The data are not publicly available due to potential copyright problems.

**Conflicts of Interest:** The authors declare no conflict of interest.

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
