# Peer review of "Residents’ Engagement Behavior in Destination Branding"

_sustainability, doi:10.3390/su14105852_

Round 1

Reviewer 1 Report

The paper is well-written, structured and easy to follow. The chosen topic is of a high interest and there is a gap in literature about destination branding and residents' engagement which the paper sufficiently addresses. The Introduction is concise and establishes the context very well highlighting the gaps the paper seeks to address. The research aim and objectives are phrased very well and in a very clear manner. 

The Literature review is comprehensive, draws on relevant sources and critical enough to demonstrate the gap in the wider academic literature. The authors have done a good job here bringing theory to practice and the hypotheses drawn are logical and well-phrased. 

I am very pleased to see separate sections about future research and practical implications. Both can be improved and expanded with more observations and comments.

Author Response

We thank the reviewer for the positive response to our work.

Reviewer 2 Report

Authors revealed an interesting research gap and presented on the basis of own research an insight to residents’ engagement behavior in destination branding.

In my opinion the main aim of the research is achieved, however more details within practical implications with reference to the results can be provided by Authors. As strengths of the paper can be indicated: clearly stated research gap, methodology and results presented in a comprehensible manner, and relatively current cited references within residents’ role and engagement behavior in destination branding. Also, the hypotheses are based on a state of research and existing literature, and clearly presented in the conceptual model with use of figure 1. Partial least squares structural equation modeling (PLS-SEM) was used by Authors to analyze the conceptual model, and the results were presented in a well-structured manner, however my knowledge is limited in this area and I cannot adequately assess the details of measurement model and structural model.

The study reveals the antecedents that affect residents’ engagement behavior in destination branding, and additionally Authors used the concepts of social identity and social exchange to discuss the results. In my opinion, a main contribution of the paper is here because the destination branding research has focused mainly on tourists and DMOs, and Authors presented residents’ perspective.

Authors stated on the basis of research that a role of residents within destination branding is crucial, for example: ‚The primary motivation for residents to support destination brand development is the improvement of the community’s social and economic well-being (447-449)’, ‚In the destination brand marketing stage, the destination marketing organization should emphasize the role of residents as brand citizens and brand ambassadors (454-454), ‚Second, the relationship between residents  and destination brands should be strengthened’ (457-458), ‚The primary motivation for residents to support destination brand development is the improvement of the community’s social and economic well-being’ (447-449). I would like to ask Authors to give some specific and concrete examples of activities/outcomes related to a role of residents as brand citizens and brand ambassadors. Maybe in the part of practical implications? What does it mean for residents ‚to support destination brand development’? On the one hand, what type of actions residents can take? On the other hand, what type of actions taken by DMOs can emphasize a role of residents as brand citizens and brand ambassadors? What precisely should be done to strengthen a relationship between resident and destination brand? Are there any examples of these actions in Zhouzhuang or - more generally - Suzhou or China which can be presented briefly in the paper? Examples/outcomes will be useful for better understanding the discussed process of residents’ engagement behavior in destination branding as well as the above-mentioned role of residents as brand citizens and brand ambassadors. Even one paragraph will be enough to make this issues clearer.

Other remarks:

Lines 72-76 - these paragraphs should be part of the manuscript or are rather guidelines for authors?

Author Response

We thank the reviewer for the positive response to our work as well as for the constructive feedback. We hope our revisions further enhance his/her opinion of the manuscript. According to the reviewer’s suggestion, we have added discussion on practical applications of the research results. Constructive suggestions help us highlight the main contributions of the study. The revised parts of the manuscript are colored in blue.

- In the literature review section, from line 72 to line 86, it's a mistake. We have removed the content.

Line 449-459

The behavior of residents' brand citizens has the characteristics of spontaneity and consciousness, and the target is mainly tourists. It includes the dimensions of willingness to help, brand enthusiasm, and propensity for further development[79]. For example, residents take the initiative to introduce the destination brands and slogans to tourists, clarify their misunderstanding of brands and deliver positive information to tourists, and maintain brand image and reputation. Brand ambassadors have rich brand knowledge, high brand commitment and positive brand behavior. The behavior of residents' brand ambassadors includes the brand recommendations and brand building, etc. For example, "Moon Grandma" in China, can communicate with tourists in 11 languages, and spread Guilin’ destination brands through the WeChat platform etc.

Line 461-467

On the one hand, destination marketing organizations need to actively guide residents to participate in the brand construction process. On the other hand, they should help residents familiarize with brand objectives, establish destination brand identity, understand destination brand meaning, elicit positive brand responses, and, ultimately, establish a loyal, active, and intense relationship between residents and the destination brand. More importantly, destination marketing organizations need to present a way that is easy for the residents to understand and act.

Reviewer 3 Report

The study is interesting as a whole. However, I have a few suggestions:

- In the literature review section, from line 72 to line 86, I don't understand what it means. I think it's a mistake.

- In my opinion, it is interesting that the authors consider the article published in Sustainability. This work can reinforce your literature review;

Prados-Peña, M. B., Gutiérrez-Carrillo, M. L., & Barrio-García, D. (2019). The development of loyalty to earthen defensive heritage as a key factor in sustainable preventive conservation. Sustainability11(13), 3516.

. I think that the results of the contrasted hypotheses would be clearer if they were shown on the relationships of the established model. They can include a new figure with these details.

In the conclusions section and specifically in the theoretical contributions, I do not agree with this statement:
The role of residents in destination branding is largely ignored.

If the authors are sure they should justify it more. Perhaps, in my opinion, what the academic literature has not delved into is the variables presented in this study (place brand ambassadorship behavior and brand citizenship behavior); and in establishing an antecedent model of these variables, but not in the fact that there is no bibliography in which The role of residents in destination branding is largely ignored.

Author Response

Comments1:

  1. In my opinion, it is interesting that the authors consider the article published in Sustainability. Prados-Peña, M. B., Gutiérrez-Carrillo, M. L., & Barrio-García, D. (2019). The development of loyalty to earthen defensive heritage as a key factor in sustainable preventive conservation. Sustainability, 11(13), 3516.

Response from the Author(s):

We thank the reviewer for the positive response to our work as well as for the constructive feedback. We hope our revisions further enhance his/her opinion of the manuscript. The literature provided is very meaningful and enriches the content of the article. We have added the above reference.

Comments 2:

  1. In the conclusions section and specifically in the theoretical contributions, I do not agree with this statement: The role of residents in destination branding is largely ignored.

Response from the Author(s):

We thank the reviewer for the positive response to our work as well as for the constructive feedback. We agreed with the opinions of reviewer and revised the statement of the paper. Reviewer’ suggestions are constructive to improve the quality of this article, and we sincerely thank you for your suggestions. The revised parts of the manuscript are colored in blue.

Line 366-368

The academic literature on residents and destination brands has not delved into place brand ambassadorship behavior and brand citizenship behavior, and lacked an antecedent model of these variables.

Reviewer 4 Report

I really enjoyed reading this article. I think it could contribute significantly to research on destination branding. However, I think that it must be improved in some of its sections. Below you can find my suggestions.

Section 2:

  • There are some passages remaining from the template (from line 72 to line 86)
  • I would suggest extending sub-section 2.6, enriching the theoretical support for H7 and H8

Section 3:

  • I would suggest to add a table concerning the socio-demographic profile of the respondents (with frequency and percentage)

Section 4:

  • I would suggest to add a table with the results of the structural model

Section 5: 

  • I would suggest to separe the discussion from the implications, adding a specific sub-section at the beginning of the paragraph. I think that the discussion should be deepened and clarified.

Author Response

We thank the reviewer for this suggestion and have tried our best to make improvements throughout the writing process. We carefully reviewed and modified grammar, spelling, and the use of English expressions. Individuals specializing in English language editing were invited to edit the wordings. In addition, we used Editage, a leading global science communication platform, language editing to improve the language of the article.

Please see the file attached with all our answers.

Reviewer 5 Report

Dear authors:

The manuscript analyzed local residents’ engagement mechanism in the destination branding process. It is interesting that the authors distinguish between destination branding and place branding, place identity and place brand identity. As a whole, the manuscript is original and of high value. Still, there have some issues that need the authors to address

The main issue of the manuscript is the expression of English. And I think the authors have specific research ability, while the inappropriate expression influences the readability of the manuscript to some extent. Therefore, I strongly recommend the authors polish the manuscript thoroughly from an English-speaking context.

Then the introduction is not deep enough, and the logical connection between the research purpose is insufficient. Besides, the introduction lacks an eye-catching point that could attract readers at first glance.

At last, there may be a mistake on page 2, lines 72-86. The content is not related to the study.

I believe as long as the English level in paper-writing can be improved, the manuscript will be more readable.

Author Response

We thank the reviewer for this suggestion and have tried our best to make improvements throughout the writing process. We carefully reviewed and modified grammar, spelling, and the use of English expressions. Individuals specializing in English language editing were invited to edit the wordings. In addition, we used Editage, a leading global science communication platform, language editing to improve the language of the article.

Thank you. We improved the introduction. The revised parts of the manuscript are colored in blue.

Line 53-57

For a long time, the pursuit of economic interests and the resulting attention to the interests of tourists make destination branding mainly focus on the external orientation. In addition, residents are the most affected stakeholders in destination branding. Res-idents can play a diversified role in destination branding, such as spreading positive word of mouth.

- In the literature review section, from line 72 to line 86, it's a mistake. We have removed the content.

Round 2

Reviewer 4 Report

Your paper has improved. I think there is still one change to be made, regarding table 5. In that table, with reference to each tested relationship, you should specify at least the beta, the t value and the p value. If you do not specify the p value, you should use the asterisks and make explicit at the bottom of the table which values of the p value are associated with the asterisks: e.g. *** p < 0.001 and so on.

Author Response

We thank the reviewer for the positive response to our work as well as for the constructive feedback. According to the reviewer’s suggestion, We have added the t value and the p value in Table 5. Constructive suggestions help us highlight the main contributions of the study. 
